# Biocomposite Materials Based on Poly(3-hydroxybutyrate) and Chitosan: A Review

**DOI:** 10.3390/polym14245549

**Published:** 2022-12-18

**Authors:** Yuliya Zhuikova, Vsevolod Zhuikov, Valery Varlamov

**Affiliations:** Research Center of Biotechnology of the Russian Academy of Sciences 33, Bld. 2 Leninsky Ave, Moscow 119071, Russia

**Keywords:** biopolymers, chitosan, polyhydroxyalkanoates (PHA), poly(3-hydroxybutyrate) (PHB), electrospinning, composites, blends, biomedicine

## Abstract

One of the important directions in the development of modern medical devices is the search and creation of new materials, both synthetic and natural, which can be more effective in their properties than previously used materials. Traditional materials such as metals, ceramics, and synthetic polymers used in medicine have certain drawbacks, such as insufficient biocompatibility and the emergence of an immune response from the body. Natural biopolymers have found applications in various fields of biology and medicine because they demonstrate a wide range of biological activity, biodegradability, and accessibility. This review first described the properties of the two most promising biopolymers belonging to the classes of polyhydroxyalkanoates and polysaccharides—polyhydroxybutyrate and chitosan. However, homopolymers also have some disadvantages, overcome which becomes possible by creating polymer composites. The article presents the existing methods of creating a composite of two polymers: copolymerization, electrospinning, and different ways of mixing, with a description of the properties of the resulting compositions. The development of polymer composites is a promising field of material sciences, which allows, based on the combination of existing substances, to develop of materials with significantly improved properties or to modify of the properties of each of their constituent components.

## 1. Introduction

Biopolymers are macromolecular compounds that are part of living organisms and are products of their vital activity.

Biopolymers are obtained from such biological sources as insects, crustaceans, and various microorganisms. Significant interest in biopolymers is associated both with the global attention to environmental pollution and with the presence of the critical advantages of biological polymers over synthetic ones. This eventually led to the commercialization of biopolymers and related products in various fields of biomedicine, ecology, and bioengineering [1,2,3,4]. The fact that biopolymers’ breakdown byproducts are not harmful to the environment is one of their distinguishing characteristics [5].

Based on the fact that biopolymers are obtained naturally from the ecosystem, they have higher economic value and biodegradability [6]. Biopolymers can be both natural (synthesized in living organisms) and synthetic (synthesized under artificial conditions) (Figure 1). Polyesters are complex organic compounds with repeating ester bonds in their composition. Polyesters produced by microorganisms are biodegradable. However, most synthetic polyesters are not biodegradable. Polyhydroxyalkanoates, polylactide, polyglycolide, and polycaprolactone are the most famous polyesters. Polyesters are widely used in the production of drug delivery systems and the manufacture of materials for biomedical applications [7]. Polysaccharides are long-chain polymeric carbohydrates consisting of monosaccharide units bound together by glycosidic bonds. This carbohydrate can react with water (hydrolysis), which produces constituent sugars (monomeric saccharides) [8,9,10]. Cellulose and chitin are the most famous polysaccharides.

This review is devoted to the features of obtaining and studying the properties of materials based on polysaccharides and polyester. As noted above, polymers of natural and microbiological origin are considered promising alternatives to synthetic biopolymers in fields ranging from packaging materials to biomedicine. However, their use is often limited by certain disadvantages of each class of biopolymers. For example, high sensitivity to water and low mechanical and thermal stability are the main limitations for the industrial application of chitosan [12,13]. In addition, high solubility reduces its barrier properties, which can lead to the complete solubility of chitosan in food products. Blending chitosan with a hydrophobic polymer can minimize solubility problems [14].

The creation of copolymers and composites is intended to eliminate some of the shortcomings of homopolymers, such as poor mechanical characteristics, low strength, and insufficient biodegradability. Overcoming disadvantages is achieved by synthesizing copolymers or creating composites with other materials, both natural and synthetic [7,15,16,17].

Typically, composites are made by different methods, such as molding, extrusion, solvent casting, electrospinning, and intercalation [18,19]. The choice of a composite manufacturing method plays a decisive role in the mechanical and physical properties of polymeric biocomposites. For example, biocomposites based on banana fiber and polylactic acid have been made using three different processing techniques, namely direct injection molding (DIM), extrusion injection molding (EIM), and extrusion compression molding (ECM), to obtain better properties [20]. Biopolymer properties are also crucial for understanding the structural complexity of biocomposites [20,21,22].

The main problem faced by researchers of polyhydroxyalkanoates and chitosan composites is related to their different natures. The review presents data on the use of different methods of composite fabrication and investigations of their properties.

## 2. Biomaterials Based on Chitosan and PHB

### 2.1. Chitosan Characteristics and Applications

Chitosan is a deacetylated derivative of chitin, mainly obtained from the exoskeletons of crustaceans [23,24,25,26]. Alternative sources of chitin are insect cuticles and fungal cell walls [27,28,29,30,31,32,33]. Chitosan is a linear heteropolysaccharide consisting of alternating units of glucosamine and N-acetyl-D-glucosamine connected by β-(1→4) glycosidic bonds, which have cationic properties: at pH ≤ 6.5, it has a positive charge (Figure 2).

Chitosan dissolves in aqueous solutions of acids. Chitosan salts (formate, lactate, citrate, acetate, ascorbate, etc.) are soluble in water [27]. Chitosan is characterized by such parameters as the degree of deacetylation and molecular weight [34], as well as the polydispersity index. These parameters affect many of the physicochemical and biological properties of chitosan, such as solubility, hydrophilicity, crystallinity, and cell affinity. At a pH value below 4, the amino groups of chitosan are protonated, which leads to electrostatic repulsion between the charged groups and swelling of the polymer. Free amino groups form intermolecular hydrogen bonds with the oxygen of neighboring chains. Amino groups make chitosan a cationic polyelectrolyte (pKa ≈ 6.5), one of the few in nature. Chitosan is protonated when dissolved in an aqueous acidic medium at pH < 6.5, but when dissolved, it has a high positive charge on the -NH3 + groups and the resulting soluble polysaccharide is positively charged. Chitosan aggregates polyanionic compounds. Chitosan has functional groups in its structure, which can be subjected to structural modification by chemical, radiation, and enzymatic methods in order to acquire new properties [34,35].

Solutions of chitosan with a high molecular weight have a high viscosity, which may limit the use of chitosan in some industries. Therefore, the process of depolymerization is widespread, with the formation of low molecular weight chitosan and oligosaccharides [36,37], which have a wide application potential [38]. Thus, chitosan with a molecular weight of 10 kDa has high antibacterial activity [39], increases disease resistance of plants [40], and demonstrates antioxidant activity [41]. Studies have shown that oligosaccharides [42,43] have lower viscosity, good absorbability, and solubility in water and under physiological conditions. Chitooligosaccharides demonstrated antimicrobial activity against a wide range of gram-positive (*M. luteus*, *S. mutans*, *S. faecalis*, *S. epidermis*, *S. aureus*, *B. subtilis*, *B. cereus*, *L. plantarum*, *B. bifidum*) and gram-negative (*E. coli*, *V. parahaemolyticus*, *V. Vulnificus*, *S. typhimurium*, *P. aeruginosa*) bacteria, as well as fungi (*Saccharomyces cerevisiae*, *Aspergillus niger*, *Candida*) [44,45,46]. Some researchers also report the effectiveness of oligosaccharides as antitumor agents, as well as providing an anti-inflammatory effect [47]. One of the areas of chitosan research is the development of new drug delivery systems, including those that have an antitumor effect. Thus, a stearic acid and chitooligosaccharide derivative was suggested as a carrier for anticancer agent intracellular transport and were effective against cells from breast cancer, liver cancer, and lung cancer [48].

The degradation of chitosan-based materials in the body leads to the formation of non-toxic amino sugars such as glucosamine or N-acetylglucosamine, which are completely absorbed by the human body. This allows considering it a promising candidate for a wide range of biomedical applications [49]. A significant number of studies have shown that chitosan has antibacterial properties [50,51,52], fungicidal [53], antitumor [54], absorbent, mucoadhesive[55,56], wound healing [57,58], hemostatic [59] properties.

Due to its polycationic properties, chitosan can interact with the surface of the cell membrane and be used as a material for bone tissue regeneration [60]. Chen et al. [61] developed a polymer stent made from chitosan films crosslinked with genipin to improve mechanical properties. The results showed not only improvement in mechanical properties but also reendothelialization of the implanted vascular stent.

The mechanical properties of chitosan have been widely studied. The authors [62] conclude that the ultimate tensile strength of chitosan fibrils is in the range of 121.5–308 MPa and Young’s modulus is 7.9–22.7 GPa. The mechanical properties of chitosan fibers are determined by the molecular weight and degree of deacetylation, the solvent used, and the source of production. Furthermore, when the relative humidity is reduced from 93% to 11%, Young’s tensile modulus of chitosan films increases from 10.9 ± 1.2 GPa to 18.8 ± 1.5 GPa [63]. The micromechanical properties of chitosan are also investigated using nanoindentation [64], with Young’s modulus ranging from 1 to 3 GPa, which correlates with the results of [65].

The study [66] is devoted to the study of chitosan films stored under controlled conditions and the change of their properties over time. It is demonstrated that the films undergo significant changes in properties during storage due to changes in the structure associated with the Maillard reaction. The rearrangement of polymer chains during storage caused structural changes, changes in mechanical properties, changes in resistance to ultraviolet and visible light, and changes in hydrophobicity. Thus, during storage from 0 to 90 days, the tensile strength of low molecular weight chitosan changed from 55 to 76 MPa and that of high molecular weight chitosan from 61 to 76 MPa. Elongation at break values decreased. This means that the films became stiffer and less tensile, which can be explained by the loss of bound water. The authors conclude, however, that the functional properties of the chitosan films remained acceptable even after 90 days of storage. The use of natural or synthetic plasticizers [67] can improve the properties of chitosan films with retention for up to 10 months. Currently, chitosan is used to create products such as films, coatings, hydro- and cryogels, micro- and nanoparticles, and matrices used for the fabrication of wound healing agents [68,69], targeted delivery of medicines [70,71,72,73] and for ophthalmology [69,74,75]. Chitosan has the characteristics that are necessary for an ideal contact lens, such as optical transparency, mechanical stability, sufficient optical correction, gas permeability, wettability, and immunological compatibility [76,77]. It’s a well-known immunomodulator [78]. Chitosan exhibits antitumor properties by inhibiting the growth of tumor cells due to the proliferation of cytolytic T-lymphocytes [44]. There are studies reporting its antiviral activity [79,80].

Mucoadhesive properties of chitosan are known, which appear due to electrostatic interactions with negatively charged epithelial membranes [81]. Thus, an improvement in the efficiency of drug delivery through the mucous membranes under the influence of chitosan was found, as well as prospects for its use in ophthalmology. [82]. A scientific study [83] describes biocompatible systems of chitosan nanoparticles with medicinal substances included in them, which can be retained on the ocular surface for a long time, penetrate the blood-ocular barrier, concentrate inside the eye, and also have a pronounced therapeutic effect.

Bioabsorbable suture materials based on chitosan are made. In work [84], a quaternized chitosan derivative was used for application to the Vicryl surface. The resulting material exhibited antibacterial activity and good cytocompatibility with human fibroblast cells. A significant number of studies report the ability of chitosan and its derivatives to accelerate wound healing [85] by modulating the secretion of enzymes and cytokines. Such materials are available in the form of electrospun fibers [86], hydrogels [87,88], membranes [89,90], films [91], 3D scaffolds [92], and sponges [85].

One of the primary issues with materials based on unmodified chitosan is that they have poor mechanical qualities. This issue is resolved by functionalizing chitosan to produce chitosan derivatives, as well as by creating composites and mixes based on chitosan and other materials.

### 2.2. Poly(3-hydroxybutyrate): Preparation, Structure, Applications

Poly(3-hydroxybutyrate) (PHB) is one of the best-known polyhydroxyalkanoates (PHA). Polyhydroxyalkanoates are a family of biodegradable polyethers typically produced microbiologically using the highly efficient producer strain *Azotobacter chroococcum* 7B (Figure 3).

PHAs are widely used in biomedicine [94] to create suture threads, wound healing materials, orthopedic pins, stents, devices for targeted tissue repair/regeneration, joint cartilage, bone implants, nerve fiber connections, etc. Although PHB has several advantages, it also has a few limitations that restrict its biological uses, including high crystallinity and brittleness.

A variety of microorganisms use polyhydroxyalkanoates as an intracellular energy source and carbon source [95]. Only prokaryotes can synthesize this type of polymer. Some PHA-synthesizing microorganisms are presented in Table 1. Most microorganisms are capable of accumulating PHA from 30% to 80% of the cell’s dry weight.

Poly(3-hydroxybutyrate) is a linear, isotactic polymer that can have a high molecular weight of up to about 3000 kDa. The conformational structure of PHB is a right-handed helix with a double helical axis. Helical conformation is stabilized by carbonyl/methyl interactions and is one of the few exceptions in nature in which its formation and stability do not depend on hydrogen bonds.

The physical and mechanical properties of PHAs, such as stiffness, brittleness, melting point, glass transition point, or resistance to organic solvents, can vary significantly, depending on the monomer composition [115]. For example, increasing the content of 3-hydroxyvalerate (HV) residues in the poly(3-hydroxybutyrate-co-3-hydroxyvalerate) (PHBV) copolymer leads to a decrease in the melting point from 180 °C for pure PHA to ~75 °C for the copolymer with 30–40 mol % HV content. There is also evidence that PHA isolated from P. oleovorans is able to dissolve in acetone, while the homopolymer of PHB shows very low solubility in acetone.

The crystalline structure of PHB is usually in the form of lamellae. The lamellae thickness varies from 4 to 10 nm depending on the molecular weight, solvent, and crystallization temperature [116,117].

The single crystal structure of PHB is a monolamellar system [118]. However, such products as films are usually multilamellar systems that assemble into multi-oriented lamellar crystals. In 3D-structures, such as scaffolds, PHB chains usually form spherulites [119]. In spherulites, lamellar PHB crystals grow radially stacked. Because of the tendency of lamellar crystals to twist, PHB spherulites usually have a banded texture. The periodicity and regularity of such twisted structures depend on both the molecular weight and the crystallization temperature of the polymer. The growth kinetics of PHB spherulites was investigated at various crystallization temperatures. At around 90 °C, the maximum crystallization volume was observed. The overall crystallization rate of PHB is maximal in the temperature range of 50–60 °C [108].

The molecular weight of poly(3-hydroxybutyrate) synthesized by wild-type bacteria ranges from 1 × 10^4^ to 3 × 10^6^ g/mol with a degree of polydispersity ~2 [93,98,119]. The glass transition temperature of PHB is ~4 °C, while the melting point is ~180 °C [120,121]. A bifurcated peak of melting temperature is also sometimes observed in homopolymers. This phenomenon can be explained by the presence of crystallites of different degrees of perfection, which can include the thermal prehistory of the sample and the broad molecular weight distribution. The densities of the crystalline and amorphous components of PHB are 1.26 and 1.18 g/cm^3^, respectively. The mechanical properties of PHB, such as Young’s modulus (~3.5 GPa), and tensile strength (~43 MPa), are similar to those of isotactic polypropylene. However, the elongation at break (5%) is much lower than that of polypropylene (400%). Consequently, PHB is a stiffer and more brittle plastic compared to polypropylene.

## 3. Formation of Composites Based on Chitosan and Poly(3-hydroxybutyrate)

Composites are multicomponent materials, usually consisting of a base (matrix) reinforced with fillers. The combination of dissimilar substances leads to the creation of a new material, the properties of which differ from the properties of each of the components. The creation of composites with hydrophilic and biocompatible polymers is one of the strategies for changing the surface properties and improving the biocompatibility of polyhydroxyalkanoates [122].

Composite materials based on PHB with the addition of chitosan will be more hydrophilic than materials based on pure PHB. This means that they will be more biocompatible and promising for creating tissue-engineered biomedical structures. It is also assumed that mixing chitosan with PHB will be able to provide a variety of primary amino groups for further modifications and thus diversify the possibilities of using such a blended material [123]. The bioactivity, osteoinductive, and osteoconductive capabilities of mixed compositions can be improved by including hydroxyapatite in their structure [124,125,126,127].

The coating for bioceramic scaffolds, consisting of Chitosan and PHB, with the addition of multi-walled carbon nanotubes, made it possible to obtain thermostable materials with good mechanical properties for tissue engineering [128]. Previously, the authors demonstrated that, in addition to PHB, coating nano-bioglass/TiO_2_ scaffolds with chitosan made it possible to achieve a decrease in the contact angle from 84° to 42°, compressive strength of the scaffolds increased from 0.01 MPa for nBG/nTiO_2_ to 0.19 MPa for nBG/nTiO_2_ with PHB/Cs, improve biological activity, and cell proliferation [129]. However, such coatings were rapidly destroyed and tended to swell.

Chitin and chitosan have also been used to improve the mechanical and thermal properties of PHB and its copolymers. Chemical modification of chitin was carried out to increase the hydrophobicity (increase in contact angle from 33° for chitin and up to 70° for composite). The crystallization temperature is 22 °C lower in the composite compared to pure PHBV. The melting temperature of PHBV increased from 154.5 °C to 165 °C for composite. [130]. Films based on PHBV/acetylated chitin nanocrystals were produced by solution casting [131]. Studies of mechanical properties showed that the ultimate tensile strength and Young’s modulus of PHBV are improved by about 24% and 43%, respectively. The contact angle increased from 31° for chitin to 68° for PHBV/chitin, and the Tc of composites is 5 °C higher than that of PHBV [131].

In addition, the melting technique cannot be applied because chitosan has a high melting point at which PHB will decompose. Additionally, the melting approach cannot be used because PHB would start to break down before chitosan melts due to its high melting point [132].

Using a micro compounder, mixtures were prepared as described in the article [133] at a temperature of 175 °C and a screw speed of 100 rpm. Initially, PHB was melted; then chitosan (5, 10, 20, 30, and 40 wt %) was introduced into the mixer. The granular samples were then subjected to injection molding. When studying the thermal properties of the obtained composites, it was shown that the inclusion of chitosan in PHB increases the glass transition temperature with a decrease in the melting temperature and crystallinity. The addition of 10 wt.% chitosan reduced the percentage of crystallinity of the chitosan/PHB composition. At the same time, a further increase in the concentration of chitosan did not cause significant changes. This is because the presence of very rigid chitosan molecules surrounding the PHB molecules made the PHB molecules in the composites inflexible and induced insufficient crystallization compared to pure PHB. Adding more chitosan reduced the thermal stability of the composites.

### 3.1. Copolymerization

A large number of studies confirm that chitosan, as well as its derivatives, has an antibacterial effect due to its ability to disrupt the normal functions of the bacterial cell membrane due to the reaction between the positive charges of chitosan and the negatively charged bacterial cell walls [134]. Usually, chitosan exhibits its antimicrobial activity in an acidic environment due to its poor solubility and the absence of protonated amino groups at pH 6.5 and above, which limits its use. One approach to overcoming some of the disadvantages of primary chitosan is to modify it with suitable functional groups. Graft copolymerization is a promising method for producing new types of hybrid materials based on chitosan with improved properties, thereby broadening its application in biomedicine and environmental protection [135,136]. This approach allows the formation of functional derivatives by covalent binding of the grafted molecule to the main chain of chitosan (Figure 4) [137].

Polyhydroxyalkanoates can be used to modify chitosan. Thus, in the study by Arslan et al. [138], poly-(3-hydroxyoctanoate), as well as PHBV and linoleic acid, were grafted to chitosan via condensation reactions between carboxyl and amino groups, while the percentage of grafted polyester varied from 7 to 52% wt., depending on the molecular weight. It has been shown that the grafting percentage depends on the molecular weight, the structure of the grafted PHAs (steric effect), and, finally, the solubility of the polymers in the polymerization medium. So, the solubility of the grafted chitosan-g-PHA copolymer can be controlled by adjusting the grafting percentage.

In another study, chitosan derivatives formed viscous solutions in water. Although the original polymer is hydrophobic, its graft derivative exhibits amphiphilic behavior in which the degree of solubility is controlled by the percentage of graft. As a result, PHB-g-chitosan copolymers are obtained, which have strong elastic films with a low melting point. Due to their biocompatibility, amphiphilic behavior, and antimicrobial activity (Table 2), polymer grafts have good potential for medical applications such as tissue engineering and drug delivery systems [139].

The compatibility of chitosan and polyhydroxyalkanoates can be increased by functionalizing polymer chains by grafting functional groups or comonomers [140]. Vernaez et al. used maleic anhydride to modify PHBV using a Brabender measuring mixer. PHBV with a cross-linking agent concentration of 10% wt., similarly mixed with chitosan powder. It is concluded that functionalization with maleic anhydride effectively increases the compatibility of the polyester matrix with chitosan particles.

In work by Hu et al., acrylic acid and chitosan, onto which hyaluronic acid was immobilized, were sutured to ozone-treated PHBV membranes [141]. These PHBV/acrylic acid membranes, which were esterified to chitosan or chitooligosaccharides to increase hydrophilicity, had antibacterial properties and improved fibroblast cell attachment.

### 3.2. Electrospinning

The electrospinning process has shown significant potential for creating fibrous scaffolds that can mimic the structure of the extracellular matrix in natural tissues (Figure 5) [142,143]. Electrospun fiber scaffolds have improved structural properties, including greater surface area and higher porosity with interconnected pores that promote cell growth and nutrient exchange [144]. Many natural and synthetic polymers, including PHB and chitosan, have been investigated for the manufacture of fibrous materials [145]. In this study, PHB was successfully blended with chitosan using TFA as a co-solvent, and the blended solution was electrospun to fabricate fibrous scaffolds for cartilage engineering. This study showed that the addition of chitosan could increase the hydrophilicity and weight loss rate (and percentage) for PHB scaffolds while maintaining the mechanical properties in a suitable range. The results obtained indicate the great potential of fibrous scaffolds made from a mixture of PHB/chitosan for further additional studies in vitro and in vivo.

The method of fiber formation using electrospinning was used in [146]. The aim of this study was to determine the effect of a polyhydroxybutyrate/chitosan/nanobioglass nanofiber scaffold fabricated by electrospinning on the proliferation and differentiation of stem cells obtained from human exfoliated deciduous teeth into odontoblast-like cells. Trifluoroacetic acid was used as a solvent; first, poly(3-hydroxybutyrate) was dissolved, and then 15 wt % chitosan and 10 wt % bioglass nanoparticles were added. Then the solution was homogenized, and electrospinning was performed. According to the results, due to the presence of chitosan and bioglass nanoparticles, the resulting matrix had good cell viability and uniform properties and was suitable for pulp capping (Figure 6).

Peripheral nerve conductors can be constructed based on biodegradable polymers. Thus, in the article by Zhou et al., biocomposite was constructed using electrospinning based on chitosan and PHB [147]. In this case, the addition of chitosan made it possible to improve wettability. The results of TGA and DSC showed that PHB/chitosan polymers could be mixed in one phase with trifluoroacetic solvent in all compositions. Furthermore, a decrease in decomposition temperature (from 286.9 to 229.9 °C) and crystallinity (from 81.0% to 52.1%) with increasing chitosan content was demonstrated. The authors conclude that PHB/chitosan fibers with different percentages can be used as medical materials with a controlled degradation rate.

Electrospinning was used to create chitosan/PHBV nanofibers [148] with various PHBV/chitosan ratios (4:1 and 2:3, respectively) as cytocompatible degradable scaffolds. Such materials ensured the proliferation and differentiation of L929 fibroblast cells, as well as a controlled rate of biodegradation [149]. PHBV/chitosan at a ratio of 4:1 demonstrated the best cell proliferation (53 ± 9% compared to PHBV/C [2:3] (32 ± 9%)) and wound healing rate (smaller wound areas (26 ± 11%), compared to control (59 ± 17%).) in vivo.

In another study, the authors developed a material based on PHB-chitosan with TFA as a solvent [150]. Aluminum oxide nanowires were added as a reinforcing phase, and 3D scaffolds were obtained by the electrospinning method. Chitosan addition reduces the tensile strength from 2.81 ± 0.15 MPa to 0.89 ± 0.26 MPa and the modulus from 126.3 ± 22.2 to 44.6 ± 0.2 MPa. The alumina provided optimum mechanical properties (11.18 ± 1.24 MPa) and increased surface roughness (492.6 ± 67 nm against 346.2 ± 23 nm for PHB-CTS ) as well as good biological properties in vitro.

Thus, composite electrospun matrices are promising materials, primarily for biomedicine. This is due to the possibility of creating a material with certain predetermined characteristics and the ease of its functionalization and modification. However here, it is important to note one problem, which is usually characteristic of materials that are difficult to mix under normal conditions. Namely, rather little is known about the homogeneity and distribution of components in such mixed materials, i.e., whether the components are combined into a single fiber or are independent networks [151]. These issues require further investigation in the creation of electroformed PHB-chitosan composites.

### 3.3. Blending Polymer Solutions in Different Solvents

Blending is a simple and effective way to obtain biomaterials with the desired properties. However, blends are often developed from the same class of biomaterials. At the same time, insufficient attention is paid to the creation of mixtures based on hydrophilic biopolymers and hydrophobic polyesters. The creation of composites and mixtures based on chitosan and PHB is associated with certain difficulties, which include selecting the optimal solvent suitable for chitosan and PHB. As mentioned above, new types of biomaterials based on PHB, and chitosan can be obtained using various mixing methods (Figure 7).

It is expected that PHB/chitosan mixtures will have good biocompatibility and high biodegradability, so they can be used as biomedical materials. The authors [152] note that the properties of mixtures of chitosan and PHA depend on the method of their preparation:

1. A method of casting polymer solutions. By casting from a PHB/chitosan mixture solution, compositions with different polymer ratios can be prepared, which means that the mixtures will have the properties of both components [153]. The main advantage of the method of mixing by casting from polymer solutions is its simplicity. Thus, to prepare composites, chitosan and PHB were dissolved in 1,1,1,3,3,3-hexafluoro-2-propanol, and the resulting mixture was poured onto a Teflon cup [154]. The introduction of chitosan reduces the crystallization of PHB from 65% to ~35% for PHB/CS 50:50 due to the harsh environment of chitosan and the formation of intermolecular hydrogen bonds between the components.

2. Blending Precipitation method [155]. In this case, PHB or its copolymers were dissolved in DMSO and mixed with a solution of chitosan in acetic acid with DMSO. Then the mixture was precipitated in excess of acetone. The precipitate was filtered and dried. The pre-deposition mixing method can be used as an effective way to improve the plasticity of chitosan by manipulating the production conditions and composition [156].

It was shown that hydrogen bonds formed between the components of the mixture, and the addition of chitosan led to a decrease in the degree of PHB crystallinity. Similar results were obtained by Khasanah et al. [157] in the study of PHB/chitin films. The crystallinity of PHB decreased (22% for PHB/CS 50:50) in mixtures compared to the pure polymer (77%), and new intermolecular hydrogen bonds formed between the CO groups of PHB, and the O–H, N–H groups of chitin. At the same time, a decrease in the degree of crystallinity makes it possible to improve the physical properties of PHB, which means expanding the potential for its practical application [154,158].

Propylene carbonate and acetic acid were used as solvents, respectively, for PHB and chitosan and further mixing of polymer solutions [159]. This made it possible to avoid the use of toxic fluorinated co-solvents. The same volumes of solutions with different concentrations of polymers were used. After mixing, acetone was added to precipitate the polymers, which were then washed, frozen, and lyophilized. As a result, composite scaffolds (chitosan-PHB-hydroxyapatite) were obtained.

3. The method of casting in the emulsion. Chitosan dissolved in acetic acid was mixed with a solution of PHB in chloroform. After casting into films, the mixture was neutralized with 0.5 M sodium hydroxide, washed with water, and dried. In this case, the chitosan solution should be sufficiently viscous, and the amount of PHB should not exceed 30% so that PHB drops dispersed in the chitosan matrix retain their stability [160]. 3T3 fibroblasts were used to study cytocompatibility. Fibroblasts were attached to the films and had a normal spreading morphology. Moreover, experiments on adhesion and cell proliferation showed that PHB/chitosan films had better cytocompatibility compared to pure chitosan films, probably due to better PHB biocompatibility and higher surface roughness of the films from the mixture [152].

In work, Ivantsova and co-authors [161] created a new biodegradable polymer composition for the controlled release of drugs based on PHB and chitosan with different percentages of components (10–90% wt.). In this case, PHB was used as a biodegradable component, and chitosan was added to improve the physical properties and the possibility of further modification due to the presence of reactive amino groups in the composition. The components were mixed using a Brabender laboratory mixer at 150 °C for 10 min, and the resulting powder was hot pressed at 160 °C.

The authors [162] formed films based on PHA with the inclusion of various polysaccharides (chitosan, pectin, hyaluronic acid) by mixing equal amounts of polymer solutions. PHB and PHBV were dissolved in chloroform, and chitosan was dissolved in 0.1 N acetic acid at pH 5.5. The solutions were filtered and mixed. When the solvent was evaporated, the films were washed with distilled water. The resulting porous films had the flexibility and plasticity characteristic of PHB and good biocompatibility with HaCaT cells (Table 3). However, there was a slight inhibition of cell proliferation on materials containing chitosan.

According to Wenling et al., chitosan/PHB films were prepared by the emulsification/casting/evaporation method [160]. A solution of chitosan in acetic acid and a solution of PHB in chloroform were prepared separately. Scanning electron microscopy showed that PHB microspheres were formed and captured by chitosan matrices, which made the film surface rough. With an increase in the PHB content, the film surface roughness increased, and the swelling coefficient of the PHB/chitosan films decreased (9.7 ± 5.0% for PHB and 119.3 ± 4.3% for Cs/PHB 70:30). This can be explained by the hydrophobicity of PHB. This result indicates that the swelling capacity of chitosan films can be reduced by adding PHB. The stress-strain curves of the blended films were almost linear and similar to the stress-strain curves of pure chitosan films, which meant that all blended films were resilient and strong. With an increase in the amount of PHB, the elastic modulus of the films decreased (8.7 ± 1.6 MPa for Cs/PHB 90:10 to 4.9 ± 0.6 MPa for Cs/PHB 70:30), and the elongation at break increased (43.3 ± 5.9% (Cs/PHB 90:10) to (82.9 ± 9.3% CS/PHB 70:30). In addition, such films demonstrated higher tensile strength compared to chitosan films (4.0 ± 1.1 MPa for Cs/PHB 90:10 to 3.4 ± 0.5 MPa for Cs/PHB 70:30)).

Iordansky et al. investigated various issues of creating mixed compositions based on PHB and chitosan for targeted drug transport using rifampicin as an example [163,164]. At the same time, methods were proposed for creating emulsions with different percentages of biopolymers by various methods [165,166] using solutions of PHB in chloroform and chitosan in 1% acetic acid. Using SEM and DSC, the composition was shown to separate into two immiscible phases, especially in the PHB concentration range of 50–60%, when the separation of the components was observed, and the PHB globules were embedded in the chitosan matrix. The authors conclude that the formation of a heterophasic immiscible structure is not a disadvantage in the development of biodegradable materials due to improved bioavailability and the potential ability to control the rate of degradation of this polymer system.

### 3.4. Using the Same Solvent for Both Polymers

Ikejima et al. [154] reported the development of biodegradable polyester/polysaccharide blend films made from bacterial poly(3-hydroxybutyrate) with chitin and chitosan. The crystallinity of such mixtures decreased with an increase in the number of polysaccharides from ~65% to ~35%, which was shown by DSC. Using the FTIR method, this trend was confirmed since, with an increase in the number of polysaccharides, the intensity of the absorption bands of the carbonyl sites changed. It was found that the suppression of PHB crystallization occurs more strongly when mixing chitosan compared with chitin. It was found that PHB in mixtures (by 13C NMR-spectroscopy) is immersed in a “glassy” medium of the polysaccharide. The resonances of chitosan in mixtures were significantly broadened compared to those of chitin. This was explained by the formation of hydrogen bonds between the carbonyl groups of PHB and the amide -NH groups of chitin and chitosan. Crystallization and environmental biodegradability have been shown for mixtures of PHB and chitosan.

In another work by T. Ikejima and Y. Inoue, 1,1,1,3,3,3-hexafluoro-2-propanol (HFIP) was used as the solvent [167]. The films were made by casting from a solution. Dynamic mechanical, and thermal analysis showed that the thermal transition temperatures of PHB amorphous regions were similar to those of pure PHB for PHB/chitin and PHB/chitosan blends (~15 °C). It is also indicated that the studied films of mixtures were biodegradable in water.

Measurement of the contact angle of scaffolds from a mixture of PHB/chitosan dissolved in TFA showed that with an increase in the content of chitosan in the mixture, the contact angle decreases, and the hydrophilicity of the mixture increases accordingly. Moreover, with an increase in the content of chitosan, the porosity of the scaffolds increases. The rate of degradation also increases. It was shown that during 14 weeks of decomposition in a buffer solution, scaffolds containing 40% chitosan lost 40% of their mass, while pure PHB lost about 10% [132]. In addition, the authors found that an increase in the content of chitosan increased the contribution of the amorphous fraction. This result indicates the suppression of PHB crystallization when mixed with chitosan. An in vitro degradability study showed that the rate of degradation of mixed scaffolds is higher than that of pure PHB scaffolds, and the dissolution of chitosan can neutralize the acidity of PHB degradation products. Based on the obtained results, the authors conclude that the prospects for using the developed scaffolds in the field of bone and cartilage tissue engineering are good.

Cheung et al. [168] also demonstrated a decrease in the degree of crystallinity of PHB and PHBV when mixed with chitosan (Figure 8).

There is also a decrease in the melting point of the mixture compared to pure PHB. In this work, films based on chitosan/PHB and chitosan/PHBV were obtained by casting from a solution of 1,1,1,3,3,3-hexafluoro-2-propanol with the addition of 1% acetic acid. At the same time, the polymers were dissolved separately at concentrations of 10 g/l and mixed in different weight ratios (100/0, 80/20, 60/40, 50/50, 40/60, 20/80, and 0/100). Then solutions were slowly evaporated at an ambient temperature for 24 h. The authors conclude that there is an intermolecular interaction between chitosan and PHB or PHBV in all compositions.

It should be noted that TFA and 1,1,1,3,3,3-hexafluoro-2-propanol are the most commonly used solvents for preparing compositions based on PHB and chitosan. They are highly toxic and quite expensive. Therefore, the search and study of other methods for obtaining mixtures based on chitosan and PHB remain relevant.

The researchers [169] used PHB and chitosan for the synthesis of gold nanoparticles because chitosan has a positive charge under acidic conditions and, as a result, can form conjugates with negatively charged polymers such as PHB. At the same time, a solution of chitosan in acetic acid was prepared and treated with ultrasound, and a weighted portion of PHB was added to it in a ratio of 4.16:1 under optimal conditions. The resulting conjugate was completely soluble, with chitosan acting as a reducing agent in it and PHB as a stabilizer. Previously, a similar method for the preparation of amide conjugates of partially depolymerized PHB and chitosan was described in [170] due to the interaction of terminal carboxyl groups of PHB with amino groups of chitosan and water-soluble conjugates were obtained.

In a number of papers, glacial acetic acid has been proposed as a solvent for polyhydroxybutyrate. Choonut et al. [171] heated a sample of PHB in glacial acetic acid with constant stirring to 118 °C until the sample was completely dissolved. Then, films were prepared by casting on a heated 120 °С Petri dish to accelerate the evaporation of the solvent. Similarly, PHB films were obtained in [172], and it was shown that the choice of film preparation temperature and the solvent evaporation rate is important in terms of such material properties as the degree of crystallinity, surface roughness, transparency, and mechanical characteristics. Thus, films obtained by casting at high temperatures had a lower tensile strength and deformation, a higher degree of crystallinity (78% against 60.5% in chloroform) and were also more transparent and smoother. The authors explained the increase in the crystallinity of the material by the fact that at a higher temperature in acetic acid, more thermal energy was available for the formation and growth of crystal structures and by the fact that chloroform is a more compatible solvent for PHB.

In their study Mukheem et al. obtained and characterized the antimicrobial properties of composite films based on PHB and chitosan and incorporated boron nitride nanoparticles [173]. PHB was dissolved in glacial acetic acid at a temperature of 118 °C. Then the solution was cooled, and nanoparticles were added at various mass ratios. A solution of chitosan in 1% acetic acid was added dropwise (at a ratio of PHB and chitosan 10:1, respectively). The resulting mixture was sonicated. Films were formed by casting on a glass plate heated to 80 °C.

### 3.5. Summary of the above Methods

Based on the work described above, creating new composites from two biopolymers of different natures is a non-trivial problem. Table 4 summarizes the main methods used to make composites of both PHB and chitosan.

Each method is not universal; each has its own advantages and disadvantages. For example, the future product will be expensive and difficult to manufacture, limiting its economic viability. Inconsistencies in physical properties (melting point) make it difficult to use melt extrusion methods. The use of some solvents, such as trifluoroacetic acid, is limited by their toxicity. However, the ease of making a composite by mixing it in a common solution makes it one of the most commonly used methods. Among all solvents, acetic acid occupies a separate place. It is possible to dissolve both PHB and chitosan in acetic acid. It is cheap and non-toxic, which is an advantage over other solvents.

## 4. Prospects for Applications of PHB/Chitosan Composite Materials

Most of the options for the practical application of materials based on both chitosan and poly(3-hydroxybutyrate) are associated with the biomedical field. Composite materials based on them also have prospects for application in medicine. For example, hydrogels for injection with the possibility of therapeutic loading were presented [176]. Gels based on PHB were obtained, and chitosan was added using the solvent-exchange method, which made it possible to load chitosan successfully into the PHB matrix. PHB was dissolved in chloroform at 95 °C; tetrahydrofuran was added and then immersed in methanol. Alcohol gels were extracted by soaking them in a solution of chitosan with acetic acid, which led to the formation of hydrogels. The hydrogels were lyophilized. Gels, with the addition of chitosan, showed more elastic behavior, improved compressive strength, and increased wettability. The increased hydrophilicity of the gel network may result in more efficient rebinding with water after the degradation of composite hydrogels compared to pure PHB. Drug-containing PHB/chitosan hydrogels showed excellent injectability and stability and also showed pH-driven release. The release of doxorubicin was significantly accelerated by lowering the pH of the medium from 7.46 to 4 due to the repulsion between positively charged chitosan and doxorubicin. This paves the way for the future development of a controlled drug release system in typical cancer microenvironments that are characterized by low acidity.

PHB and chitosan were used to create reservoir-type composite microparticles that could be used in the field of controlled drug delivery [177]. At the same time, PHB microspheres containing piroxicam or ketoprofen as model objects [178] were also included in chitosan matrices. The addition of chitosan crosslinked with glutaraldehyde at various concentrations makes it possible to control the release of drugs.

The potential uses of materials based on chitosan and PHB are not just limited to the biomedical industry. In the creation of biodegradable plastics, natural polymers are thought to be promising alternatives to synthetic polymers. In order to create new packaging materials that won’t pollute the environment, their use can be justifiable. Ecosystem preservation and pollution reduction are both greatly improved by the use of biopolymers [179].

Due to chitosan’s specific antibacterial activity, which demonstrates potent activity against bacteria, fungi, and yeast [180], its use in this field is justified. This makes it possible to create antimicrobial packaging materials [17]. In order to develop antimicrobial packaging, chitosan and its nanoforms have been widely used to reduce microbial growth in food products. Notably, they have been widely used to make edible coatings and edible films to increase the shelf life of food products [181]. A nanocomposite obtained by dispersing silk nanodisks in a chitosan matrix has been used as an edible coating to increase the shelf life of perishable food products [182].

Poly-3-hydroxybutyrate is used to make bioplastics because its characteristics are similar to those of typical petroleum-based polymers such as polypropylene (PP), polystyrene (PS), polyethylene (PE), and polyethylene terephthalate (PET) [183]. However, its application in the food packaging sector is underdeveloped due to the moderate barrier, thermal, and mechanical properties of this biopolymer [184]. Therefore, for this purpose, PHB is often combined with other materials, such as nanoparticles [185]. Chitosan has been used to modify PHB in the creation of food packaging [186]. Biohydrothermally synthesized ZnO-Ag nanocomposites were used as a filler to improve mechanical properties and impart additional antimicrobial properties. The resulting material showed excellent prospects for replacing non-degradable plastic for food packaging.

## 5. Conclusions

One of the strategies for improving the properties of biopolymers is the creation of new materials by obtaining biocomposites from polymers that differ in their properties. This review is devoted to the methods of creating and studying the properties and applications of composite materials based on polysaccharide chitosan and polyester PHB. Some of the disadvantages of the two classes of biomaterials, polyesters, and polysaccharides, can be overcome by blending them. For example, the incorporation of hydrophilic biopolymers such as chitosan into hydrophobic polyesters can provide functional groups for further modification, such as conjugation with growth factors, as well as reduce the crystallinity of the polymers and vary their biodegradability. In turn, the addition of polyesters to hydrophilic biopolymers can reduce their tendency to overswell. In this review, the features of obtaining biocomposites using different methods were considered: electrospinning, copolymerization, and the creation of mixed compositions by mixing polymer solutions in various solvents. Blending the two classes of biomaterials will lead to the development of some new biodegradable materials with improved properties for various applications.

## Figures and Tables

**Figure 1 polymers-14-05549-f001:**
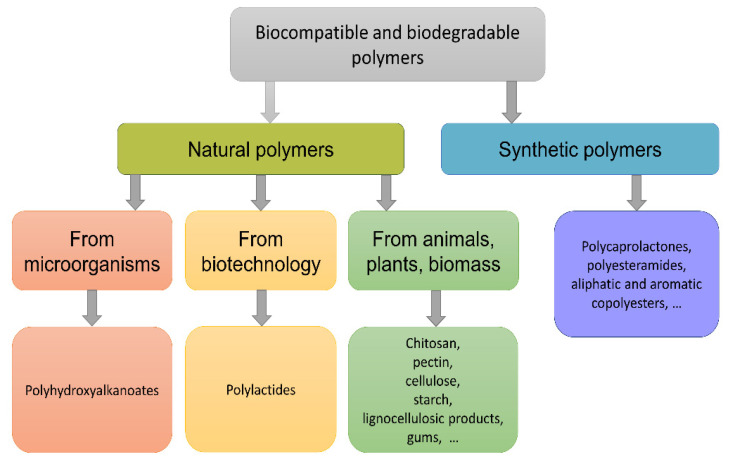
Classification of biodegradable polymers. Adapted from [11].

**Figure 2 polymers-14-05549-f002:**
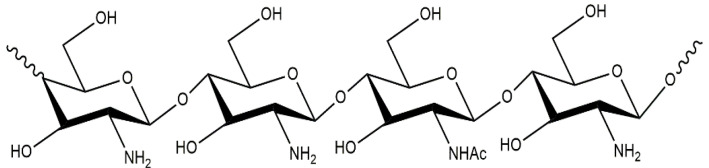
Chitosan structure.

**Figure 3 polymers-14-05549-f003:**
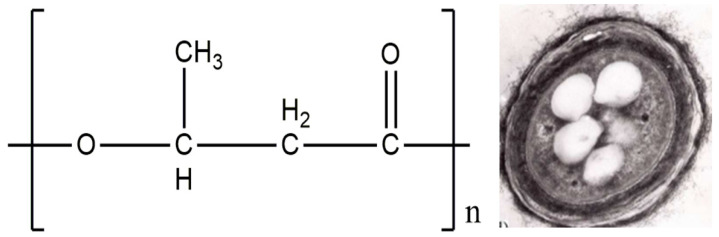
The formula of poly(3-hydroxybutyrate) and bacterial cell morphology of strain producer *A. chroococcum* during growth, reprinted with permission from [93].

**Figure 4 polymers-14-05549-f004:**
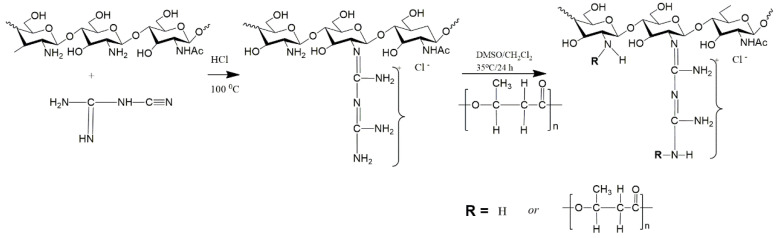
ChG−grafted PHB copolymer. Adapted from [137].

**Figure 5 polymers-14-05549-f005:**
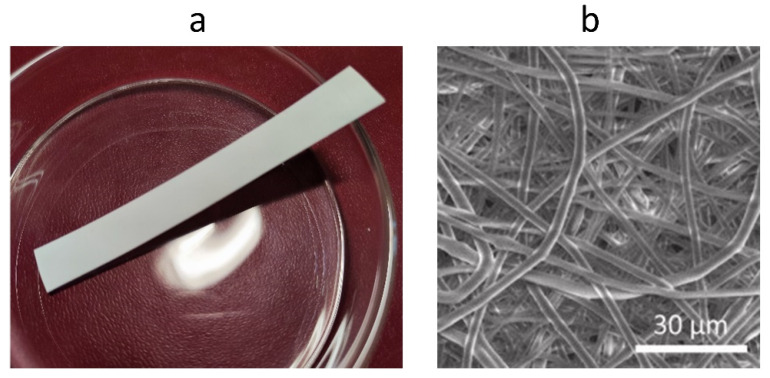
(**a**)—PHB film obtained by electrospinning; (**b**)—SEM image of PHB film, adapted with permission from [143].

**Figure 6 polymers-14-05549-f006:**
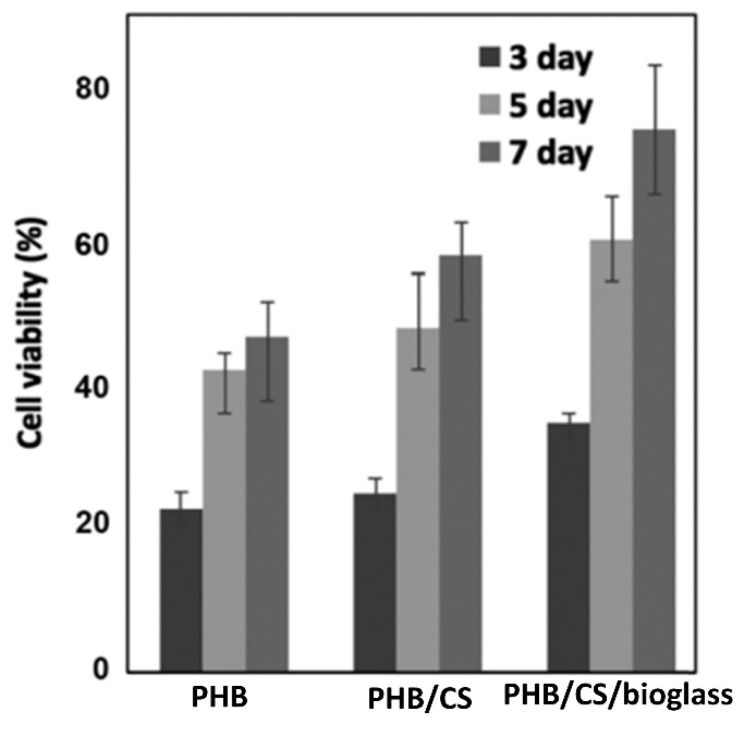
Cell viability, adapted from [146].

**Figure 7 polymers-14-05549-f007:**
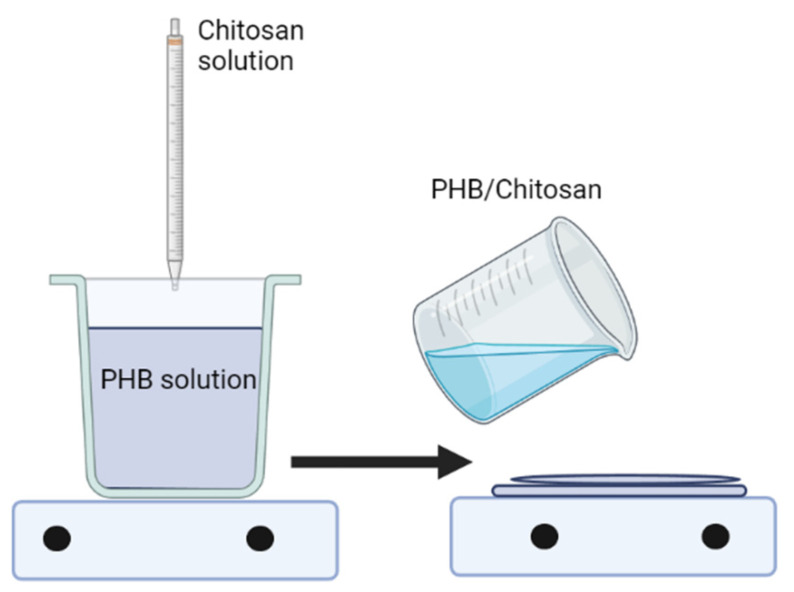
A method of casting polymer solutions.

**Figure 8 polymers-14-05549-f008:**
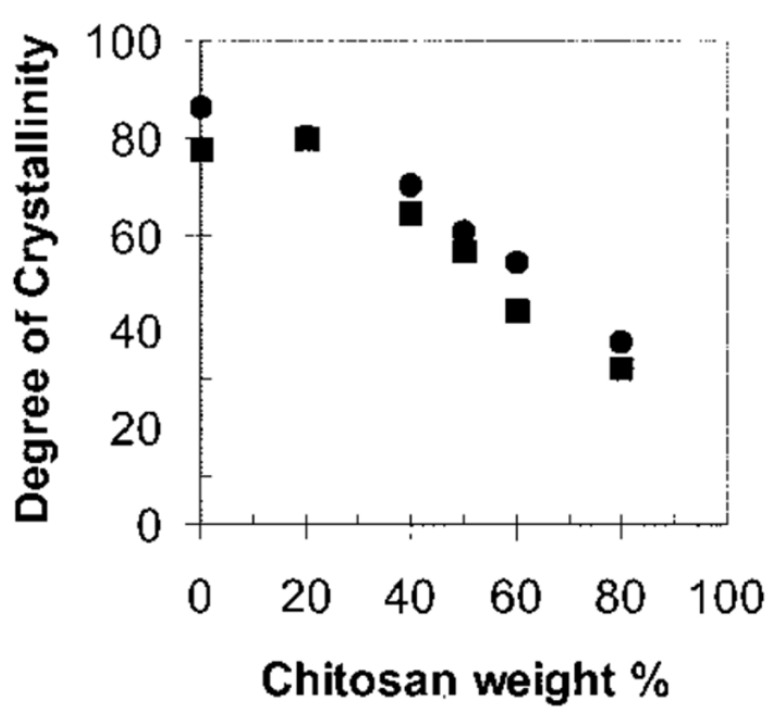
The degree of crystallinity of PHB (●) and P(HB-co-HV) (■), adapted from [168].

**Table 1 polymers-14-05549-t001:** Some representatives of prokaryotes that are capable of synthesizing PHA as an intracellular energy source.

*Acinetobacter*	[96]
*Azotobacter*	[93,97,98,99]
*Bacillus*	[100,101]
*Clostridium*	[102]
*Escherichia*	[103]
*Halobacterium*	[104]
*Methylobacterium*	[105,106]
*Micrococcus*	[107]
*Nitrobacter*	[108]
*Parapedobacter*	[109]
*Pseudomonas*	[110,111]
*Rhizobium*	[112]
*Streptomyces*	[113,114]

**Table 2 polymers-14-05549-t002:** Antimicrobial activity of ChG-g-PHB Adapted from [139].

Sample	Inhibition Zone (mm)		
	*S. pneumonia*	*E. coli*	*A. fumigatus*
ChG-g-PHB	21.30 ± 2.10	21.80 ± 2.10	19.40 ± 1.5
Ampicillin	23.80 ± 0.20	-	-
Gentamicin	-	19.90 ±0.30	-
Amphotericin	-	-	23.70 ± 0.10

**Table 3 polymers-14-05549-t003:** Comparison of the properties of PHB and PHB/Cs adapted from [162].

Sample	Young’s Modulus (MPa)	Elongation at Break (%)	Tensile Strength (MPa)	Contact Angle (Degrees)	Proliferation of HaCaT Cells (% vs. Control)
PHB	1640	1.4	12.9	84	22 ± 19
PHB/Cs	334	1.6	3.3	93	90 ± 10

**Table 4 polymers-14-05549-t004:** Advantages and disadvantages of methods for creating composites.

Methods	References	Advantages	Disadvantages
Extrusion, melt functionalization	[140,174,175]	There is no need to use expensive and toxic solvents.	Melting temperatures of PHB and chitosan are different. Thermal degradation of polymers and reduction in molecular weight can occur. Expensive specific equipment is required.
Copolymerization	[137,138,139,140]	Covalent bonding ensures the creation of a single branched structure.	Difficulty in synthesis, selection of optimal conditions is necessary, in some cases it is not easy to determine the degree of grafting
Electrospinning	[143,144,145,146,147,148,149,150]	Creating products with a variety of structures (controlled fiber size, porosity)	Requires expensive equipment. Difficult to manufacture. Limited choice of solvents
Blending in different solutions	[152,153,154,155,156,157,158,159,160,161]	The relative ease of creating the material. No specific equipment is required.	It is necessary to carefully select the mixing conditions, the ratio of polymers due to the problem of stability
Blending in a common solution	[132,154,167,168,169,170,171,172,173]	Relative stability, simplicity of manufacturing	The most widely used common solvents are expensive and toxic.

## Data Availability

Not applicable.

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
