# Peer review of "Biocomposite Materials Based on Poly(3-hydroxybutyrate) and Chitosan: A Review"

_polymers, 2022, doi:10.3390/polym14245549_

Round 1

Reviewer 1 Report

This work presents a review of the use of two biopolymers, chitosan and PHB, to create a biocomposite. The paper is focused on the existing methods to create this composite including copolymerization, electrospinning, and different ways of mixing. This is an interesting topic but some questions must be addressed before considering this paper for publication.

The main problem of the paper is that the review is focused only in a description of results published in scientific literature. The contribution of a review paper does not need to be original results but a critical analysis of the published results. In this paper there is a detailed explanation of the state of the art, but the critical analysis is not included. Section “3” includes the description of different manufacturing methods, to include a critical analysis authors should include a subsection “3.5” with a comparison of the different methods and a description of their advantages and disadvantages. A table with a summary of the comparison can also help to understand why different methods can be used.

Other minor comments:

- Section 2 describes the properties of chitosan and PHB. The mechanical properties of PHB are described (Young modulus, tensile strength…) but the mechanical properties of chitosan are not included. Are there paper that analysis the ageing of the stability of these materials? They should be included in this section.

- In the introduction authors mention that “biocomposites based on banana fiber and PLA have been made using three different methods”. Authors should mention what three methods have been used.

Author Response

We would like to thank the reviewer for constructive comments that help to improve the quality of our paper.

Here are our answers to your questions:

The main problem of the paper is that the review is focused only in a description of results published in scientific literature. The contribution of a review paper does not need to be original results but a critical analysis of the published results. In this paper there is a detailed explanation of the state of the art, but the critical analysis is not included. Section “3” includes the description of different manufacturing methods, to include a critical analysis authors should include a subsection “3.5” with a comparison of the different methods and a description of their advantages and disadvantages. A table with a summary of the comparison can also help to understand why different methods can be used.

Thank you for the suggestion. We have added section 3.5 «Summary of the above methods» on lines 574-586.

Other minor comments:

- Section 2 describes the properties of chitosan and PHB. The mechanical properties of PHB are described (Young modulus, tensile strength…) but the mechanical properties of chitosan are not included. Are there paper that analysis the ageing of the stability of these materials? They should be included in this section.

Thank you for the suggestion. Information about these properties has been added to the text of the article (lines 129-150):

The mechanical properties of chitosan have been widely studied. The authors [62] conclude that the ultimate tensile strength of chitosan fibrils is in the range of 121.5-308 MPa and the Young's modulus is 7.9-22.7 GPa. The mechanical properties of chitosan fi-bers are determined by the molecular weight and degree of deacetylation, the solvent used, and the source of production. Furthermore, when the relative humidity is reduced from 93% to 11%, the Young's tensile modulus of chitosan films increases from 10.9 ± 1.2 GPa to 18.8 ± 1.5 GPa [63]. The micromechanical properties of chitosan are also investigated using nanoindentation [64], with Young's modulus ranging from 1 to 3 GPa, which cor-relates with the results of  [65].

The study [66] is devoted to the study of chitosan films stored under controlled con-ditions and the change of their properties over time. It is demonstrated that the films un-dergo significant changes in properties during storage due to changes in structure associ-ated with the Maillard reaction. The rearrangement of polymer chains during storage caused structural changes, changes in mechanical properties, changes in resistance to ul-traviolet and visible light, and changes in hydrophobicity. Thus, during storage from 0 to 90 days, the tensile strength of low molecular weight chitosan changed from 55 to 76 MPa and that of high molecular weight chitosan from 61 to 76 MPa. Elongation at break values decreased. This means that the films became stiffer and less tensile, which can be ex-plained by the loss of bound water. The authors conclude, however, that the functional properties of the chitosan films remained acceptable even after 90 days of storage. The use of natural or synthetic plasticizers [67] can improve the properties of chitosan films with retention for up to 10 months.

- In the introduction authors mention that “biocomposites based on banana fiber and PLA have been made using three different methods”. Authors should mention what three methods have been used.

Thanks for the addition.

The composites were manufactured using three different processing techniques, namely direct injection molding (DIM), extrusion injection molding (EIM), and extrusion compression molding (ECM). https://doi.org/10.1016/j.compositesb.2019.107535. Text was added to the article on lines 69-70

Reviewer 2 Report

Biocomposite materials based on poly(3-hydroxybutyrate) and chitosan: a review
This manuscript has 2 scope, 1st is PHB and 2nd is chitosan.
One paper should focus on 1 scope. 
Thus this paper should be rejected.

Author Response

We would like to thank the reviewer for constructive comments that help to improve the quality of our paper.

Here are our answers to your questions:

This manuscript has 2 scope, 1st is PHB and 2nd is chitosan. One paper should focus on 1 scope. Thus this paper should be rejected.

Thank you for your comment. The purpose of our review is to summarize information about the creation of a biodegradable and biocompatible composite from two polymers belonging to different classes: polyhydroxybutyrate and chitosan. The most common information about this composite is brief and fragmentary, which says only about the possibility of creating this composite.
For a more complete disclosure of the topic, it is necessary first of all to characterize these two polymers, describe their physical and chemical properties, and describe their applications. The description of these polymers is given in a short, concise form since there are quite a number of reviews where only one of the presented polymers acts as a subject.
In addition, studies of composites made of several components are not uncommon. They usually also provide information about its constituent parts:

Bio-nano-composites containing at least two components, chitosan and zein, for food packaging applications: A review of the nano-composites in comparison with the conventional counterparts (https://doi.org/10.1016/j.carbpol.2021.119027)

Chitosan, alginate and other macromolecules as activated carbon immobilizing agents: A review on composite adsorbents for the removal of water contaminants (https://doi.org/10.1016/j.ijbiomac.2020.08.118)

Characterization of polyhydroxybutyrate-hydroxyvalerate (PHB-HV)/maize starch blend films (https://doi.org/10.1016/j.jfoodeng.2008.04.022)

Fabrication aspects of PLA-CaP/PLGA-CaP composites for orthopedic applications: A review (https://doi.org/10.1016/j.actbio.2012.01.031)

Thus, our review focused on one scope: the creation of a composite of polyhydroxybutyrate and chitosan, which corresponds to the reviewer's requirement.

Reviewer 3 Report

The manuscript is a review focused on preparation methods of biocomposites based on poly(3-hydroxybutyrate) and chitosan, with a description of the properties of the resulting compositions that may have interest in several application fields (e.g. biomedical, pharmaceutical, agrochemical, antimicrobial packaging). 

The topic is current and interesting and fits well into the debate on methods for improving the circularity of plastics.

The manuscript is clearly written and well organized, and comprehensively reviews the state-of-the-art on the topic. Only minor points require improvements.

In particular, the description of the properties of the reviewed compositions is often only qualitative and generic: additional quantitative data (also in form of tables, graphs and images) should be included.

In the last paragraph, the authors focus on the prospect applications of the reviewed compositions in medical field only. In the title and the elsewhere in the manuscript the considered applications are wide, not focused only on this specific sector. Therefore, I suggest to expand this paragraph by including application perspectives in other technological fields as well.

Author Response

We would like to thank the reviewer for constructive comments that help to improve the quality of our paper.

Here are our answers to your questions:

In particular, the description of the properties of the reviewed compositions is often only qualitative and generic: additional quantitative data (also in form of tables, graphs and images) should be included.

Thanks for the suggestion. Additional numerical data has been added to the text on lines: 255 -257; 261-270; 374-376; 380-384; 416; 428-429; 472; 477-481; 497; 511; 561-562

Tables 2, 3; Figures 6, 8 were added.

In the last paragraph, the authors focus on the prospect applications of the reviewed compositions in medical field only. In the title and the elsewhere in the manuscript the considered applications are wide, not focused only on this specific sector. Therefore, I suggest to expand this paragraph by including application perspectives in other technological fields as well.

Thanks for the suggestion. The following text (lines 611-634) has been inserted into the text of the article:

The potential uses of materials based on chitosan and PHB are not just limited to the biomedical industry. In the creation of biodegradable plastics, natural polymers are thought to be promising alternatives to synthetic polymers. To create new packaging materials that won't pollute the environment, their use can be justifiable. Ecosystem preservation and pollution reduction are both greatly improved by the use of biopolymers [180].

Due to chitosan's specific antibacterial activity, which demonstrates potent activity against bacteria, fungi, and yeast [181], its use in this field is justified. This makes it possible to create antimicrobial packaging materials [17]. To develop antimicrobial packaging, chitosan and its nanoforms have been widely used to reduce microbial growth in food products. Notably, they have been widely used to make edible coatings and edible films to increase the shelf life of food products [182]. A nanocomposite obtained by dispersing silk nanodisks in a chitosan matrix has been used as an edible coating to increase the shelf life of perishable food products [183].

Poly-3-hydroxybutyrate is used to make bioplastics, because its characteristics are similar to those of typical petroleum-based polymers such as polypropylene (PP), poly-styrene (PS), polyethylene (PE), and polyethylene terephthalate (PET) [184]. However, its application in the food packaging sector is underdeveloped due to the moderate barrier, thermal, and mechanical properties of this biopolymer [185]. Therefore, for this purpose, PHB is often combined with other materials, such as nanoparticles [186]. Chitosan has been used to modify PHB in the creation of food packaging [187]. Biohydrothermally synthesized ZnO-Ag nanocomposites were used as a filler to improve mechanical properties and impart additional antimicrobial properties. The resulting material showed excellent prospects for replacing non-degradable plastic for food packaging.
